

# Evaluating cloud liquid detection using cloud radar Doppler spectra in a pre-trained artificial neural network against Cloudnet liquid detection

Heike Kalesse-Los[1,2], Willi Schimmel[1], Edward Luke[3], and Patric Seifert[2]

[1]Institute for Meteorology, Universität Leipzig, Leipzig, Germany
[2]Leibniz Institute for Tropospheric Research, Leipzig, Germany
[3]Environmental and Climate Sciences Department, Brookhaven National Laboratory, Upton, New York, USA

**Correspondence:** Heike Kalesse (heike.kalesse@uni-leipzig.de)

**Abstract.** Detection of liquid-containing cloud layers in thick mixed-phase clouds or multi-layer cloud situations from ground-based remote sensing instruments still pose observational challenges yet improvements are crucial since the existence of multi-layer liquid layers in mixed-phase cloud situations influences cloud radiative effects, cloud life time, and precipitation formation processes. Hydrometeor target classifications such as Cloudnet that require a lidar signal for the classification of liquid are limited to the maximum height of lidar signal penetration and thus often lead to underestimations of liquid-containing cloud layers. Here we evaluate the Cloudnet liquid detection against the approach of Luke et al. (2010) which extracts morphological features in cloud-penetrating cloud radar Doppler spectra measurements in a artificial neural network (ANN) approach to classify liquid beyond full lidar signal attenuation based on the simulation of the two lidar parameters particle backscatter coefficient and particle depolarization ratio. We show that the ANN of Luke et al. (2010) which was trained in Arctic conditions can successfully be applied to observations in the mid-latitudes obtained during the seven-week long ACCEPT field experiment in Cabauw, the Netherlands, 2014. In a sensitivity study covering the whole duration of the ACCEPT campaign, different liquid-detection thresholds for ANN-predicted lidar variables are applied and evaluated against the Cloudnet target classification. Independent validation of the liquid mask from the standard Cloudnet target classification against the ANN-based technique is realized by comparisons to observations of microwave radiometer liquid water path, ceilometer liquid-layer base altitude, and radiosonde relative humidity. Four conclusions were drawn from the investigation: First, it was found that the threshold selection criteria of liquid-related lidar backscatter and depolarization alone control the liquid detection considerably. Second, nevertheless, all threshold values used in the ANN-framework were found to outperform the Cloudnet target classification for deep or multi-layer cloud situations where the lidar signal is fully attenuated within low liquid layers and the cloud reflectivity in higher cloud layers was sufficiently high to be detectable by the cloud radar. Third, in convective situations for which lidar data is available and for which the imprint of cloud microphysics on the radar Doppler spectrum is decreased, Cloudnet outperforms the ANN retrieval. Fourth, in high-level clouds both approaches (Cloudnet and the ANN technique), are limited.



# 1 Introduction

In mixed-phase clouds the variable mass ratio between liquid water and ice as well as its spatial distribution within the cloud plays an important role for cloud life time, precipitation processes, and the radiation budget (Sun and Shine, 1994; Yong-Sang et al., 2014; Morrison et al., 2012). The complexity of interactions in mixed-phase clouds may result in parameterizations

that are based on highly uncertain mixed-phase cloud classifications and thus lead to a misrepresentation of those clouds in models of all scales. Illingworth (2007) compared vertical ice water and liquid-water content as observed by a combination of ground-based radar, lidar, and microwave radiometer (MWR) comprised within the Cloudnet project with Global Climate Models (GCM). They showed that many GCMs underestimate the presence of mid-level clouds (As, Ac) by at least 30 % and that there is a large spread in the stated frequency of occurrence of liquid water in the models. This underestimation of the

supercooled liquid fraction (SLF) in mixed-phase clouds in many GCM was e.g. also described in Komurcu et al. (2014). Tan et al. (2016) argued that a realistic representation of the SLF in GCM is needed to better constrain the equilibrium climate sensitivity. They stated that this can only be reached by more accurate observations of the distribution of supercooled liquid in mixed-phase clouds. This remains a challenge due to the difficulty of identifying the presence of supercooled liquid water layers embedded in cloud regions dominated by ice (Shupe et al., 2008; Luke et al., 2010; Silber et al., 2020). Besides single-

layer mixed-phase clouds existing of a supercooled liquid top where ice particles are nucleated and precipitate out, multi-layer clouds (MLC) often exist (Vassel et al., 2019). MLC can interact microphysically via the seeder-feeder effect (e.g., (Cotton and Anthes, 1989; Hobbs and Rangno, 1998; Ramelli et al., 2021; Radenz et al., 2019), i.e. ice crystals nucleated in an upper liquid layer can fall into lower liquid layers, interact with its hydrometeors and influence cloud lifetime and precipitation efficiency. We thus argue that it is important to improve the detection of multi-layer liquid layer occurrences.

Synergistic measurements of cloud Doppler radar and polarization lidar can be used to identify cloud thermodynamic phase in mixed-phase clouds (e.g., Shupe (2007); Illingworth (2007); de Boer et al. (2009); Kalesse et al. (2016a) based on differences in the scattering mechanisms at the different wavelengths. While cloud radars are highly sensitive to large particles such as ice crystals (backscattering cross section is proportional to the particle size $D^6$), lidars are sensitive to higher concentrations of smaller particles such as cloud droplets and aerosol particles as the backscattering cross section is proportional to the projected

surface area of the scatterers (O'Connor et al., 2005). As an additional variable, the state of polarization of the received lidar backscatter cross section gives information about particle shape. This is usually utilized by means of the detection of the circular or linear depolarization ratio (Sassen, 1991), hereafter referred to as lidar depolarization ratio. When multiple scattering is negligible, a low (high) lidar depolarization ratio indicates the presence of spherical (non-spherical) particles (Hu et al., 2006). Except for small quasi-spherical ice particles, ice is usually non-spherical so that the lidar depolarization ratio

can also be used to infer cloud phase (Seifert et al., 2010). Concluding, liquid-dominated layers are characterized by high lidar backscattering cross section, low lidar depolarization ratio concurrent with small radar reflectivities and small mean radar Doppler velocities. Ice-dominated layers lead to a low lidar backscattering cross section, a high lidar depolarization ratio as well as higher radar reflectivities and higher mean Doppler velocities. Such synergistic lidar-radar retrievals are however only applicable up to the maximum lidar observation height determined by complete signal attenuation at a penetrated optical depth





of about three and thus do not allow for the characterization of cloud liquid in the entire vertical column, e.g. in the presence of multi-layered mixed-phase clouds.

Since cloud Doppler radars are able to penetrate multiple liquid layers, they can be used to detect warm and supercooled liquid layers (SCL) beyond the lidar measurement range via identification of morphological features in the cloud radar Doppler

spectrum (Luke et al., 2010; Verlinde et al., 2013; Kalesse et al., 2016b) and thus have great potential to characterize the distribution of SCL in the entire vertical column. Specifically, if cloud ice and liquid are observed in the same radar sampling volume and if their populations are sufficiently separated by their respective terminal fall velocities, the cloud radar Doppler spectra may contain multiple peaks. Since the terminal velocity of small cloud droplets is negligible they cause a peak at about $0 \, \mathrm{m \, s^{-1}}$ in the Doppler spectra; any deviation from this is caused by vertical motions (Shupe et al., 2004). Ice particles have

larger and broader fall velocity distributions and thus cause a spectral peak at higher Doppler velocities. If the fall velocity difference between liquid and ice is small (for example when the ice population is comprised of smaller crystals), single-peak skewed (non-Gaussian) Doppler spectra are observed (Williams et al., 2018). Sub-volume turbulence does however induce spectrum broadening which can smear microphysically-induced morphological features in the Doppler spectrum (Kollias et al., 2007). The separation of both hydrometeor populations is thus only possible if the cloud radar settings are optimized to reduce

spectrum broadening by a short dwell time, a small beam width, and a small resolution volume (Kollias et al., 2016). Sufficient range-dependent sensitivity of the cloud radar is also required as the reflectivity of the liquid peak comprised of small droplets can be as low as -40 dBz for convective situations Lamer et al. (2015).

As specific technical settings and cloud conditions are required in order to identify liquid water directly from cloud radar measurements, more sophisticated approaches are needed to make cloud radars applicable to a broader range of conditions.

Artificial neural networks (ANN) are increasingly being used in atmospheric science to evaluate large datasets and/or to combine the advantages of different sensors. In short, ANNs are mathematical models trained to recognize patterns. Validation is often done by comparison to other (physical) retrievals. As emphasized in Liljegren (2009), ANN-based retrievals have been proven to be reliable statistical techniques that are preferable to computationally expensive variational retrievals for certain applications. Liljegren (2009) developed an ANN algorithm in which G-band vapor radiometer measurements are used to retrieve

low amounts of liquid water and water vapor. Strandgren et al. (2017a) determine cirrus properties from the SEVIRI imager on Meteosat Second Generation satellites based on a set of ANN trained SEVIRI thermal observations and satellite-based lidar backscatter products among others. Andersen et al. (2017) use an ANN based on 15 years of monthly averaged Moderate Resolution Imaging Spectroradiometer (MODIS) liquid cloud products to determine the drivers of marine liquid-water cloud occurrence. All of the above studies employ multi-layer perceptrons (MLP, a specific type of feed-forward artificial neural net-

work) that are commonly used in atmospheric sciences as they are able to model highly nonlinear functions (Andersen et al., 2017). Generally speaking, a vector of output data is estimated from an input data vector by modeling the relationship between the input- and output data. The training of the MLP is done for a variety of examples where the input- and corresponding output is known. The MLP structure consists of an input layer, a chosen number of hidden layers, and an output layer. Each of these layers is made of a certain number of neurons that exchange information in a way that the output of the previous layer is used to

process the output for each connected neuron in the subsequent layer according to the corresponding numeric weights assigned





to each neuron–neuron connection through an activation function (Strandgren et al., 2017b). By using error back-propagation introduced in Rumelhart et al. (1986), the numeric weights of the neurons are adjusted in an iterative process until the squared error between the predicted (estimated) output and the known reference output data reaches its minimum.

In the present study a pre-trained (in Arctic conditions) ANN developed by Luke et al. (2010) for cloud radar-based liquid detection beyond full lidar signal attenuation is applied to mid-latitude observations (Section 2). The objective of the study is to evaluate the ANN-based liquid classification against the Cloudnet target classification (Hogan and O'Connor, 2006) by using independent measurements of MWR liquid water path (LWP), first liquid-dominated cloud base height from ceilometer observations and relative humidities with respect to liquid as obtained from radio soundings (Section 3). A short conclusion summarizing the findings is provided in Section 4.

## 2   Methods

### 2.1   Observations

#### 2.1.1   ACCEPT field experiment

Data used in this study were obtained during the Analysis of the Composition of Clouds with Extended Polarization Techniques (ACCEPT) field experiment which took place at the Cabauw Experimental Site for Atmospheric Research (CESAR, (51.971°N, 4.927°E)) in the Netherlands during 1 October- 18 November, 2014. During that field experiment, the remote-sensing instrumentation suite operated by the Royal Netherlands Meteorological Institute (KNMI) was complemented by the Leipzig Aerosol and Cloud Remote Observations System (LACROS; Büehl et al. (2013)) mainly consisting of a vertically-pointing 35 GHz MIRA-35 cloud radar (Görsdorf et al., 2015), a ceilometer, a multi-wavelength polarization Raman lidar (PollyXT; Engelmann et al. (2016)), and a HATPRO-MWR (Rose et al., 2005). Additionally, a new polarimetric hybrid-mode 35 GHz cloud radar (named hybrid MIRA-35) from METEK GmbH described in Myagkov et al. (2016a, b) and the Transportable Atmospheric Radar (TARA, S-band) operated by the TU-Delft were deployed (Pfitzenmaier et al., 2017).

#### 2.1.2   MIRA-35 characteristics

In the present study, data from the vertically-pointing MIRA-35 was used as input to the ANN of Luke et al. (2010) to predict liquid beyond full lidar signal attenuation. The MIRA-35 was operated with a pulse length of 208 ns, resulting in a vertical range resolution of 31.18 m. Incoherent averages of 20 Doppler spectra produced from a series of 256 consecutive radar pulses with a pulse repetition frequency of 5000 Hz led to a temporal resolution of 1.024 s. The MIRA-35 Doppler spectra resolution was 8 cm s$^{-1}$.

#### 2.1.3   Cloudnet target classification

The observations of the MIRA-35, the ceilometer and the MWR have been processed using the widely-used Cloudnet processing chain. One of the main products of Cloudnet is the target classification product (Hogan and O'Connor, 2006) which is





illustrated in Fig. 1 and which we use to validate the ANN-predicted liquid detections. In order to classify a cloud volume to contain liquid, the Cloudnet target classification algorithm requires a valid lidar attenuated backscatter coefficient. For deep- or multiple liquid layers and situations with low-level fog the lidar signal can get fully attenuated, so the Cloudnet target classification thus underestimates the occurrence of liquid in the entire vertical atmospheric column and overestimates the presence

5   of ice as target category (Griesche et al., 2020). Such a situation is depicted in the synergistic radar-lidar observables and the resulting Cloudnet target classification in Fig. 1. The signals of the PollyXT lidar /ceilometer were fully/partially attenuated by the near-surface fog occurring after Nov 18, 2014 07:30 UTC so that the cloud in 1.5-2.5 km around the 0 °C-isotherm was classified as ice cloud.

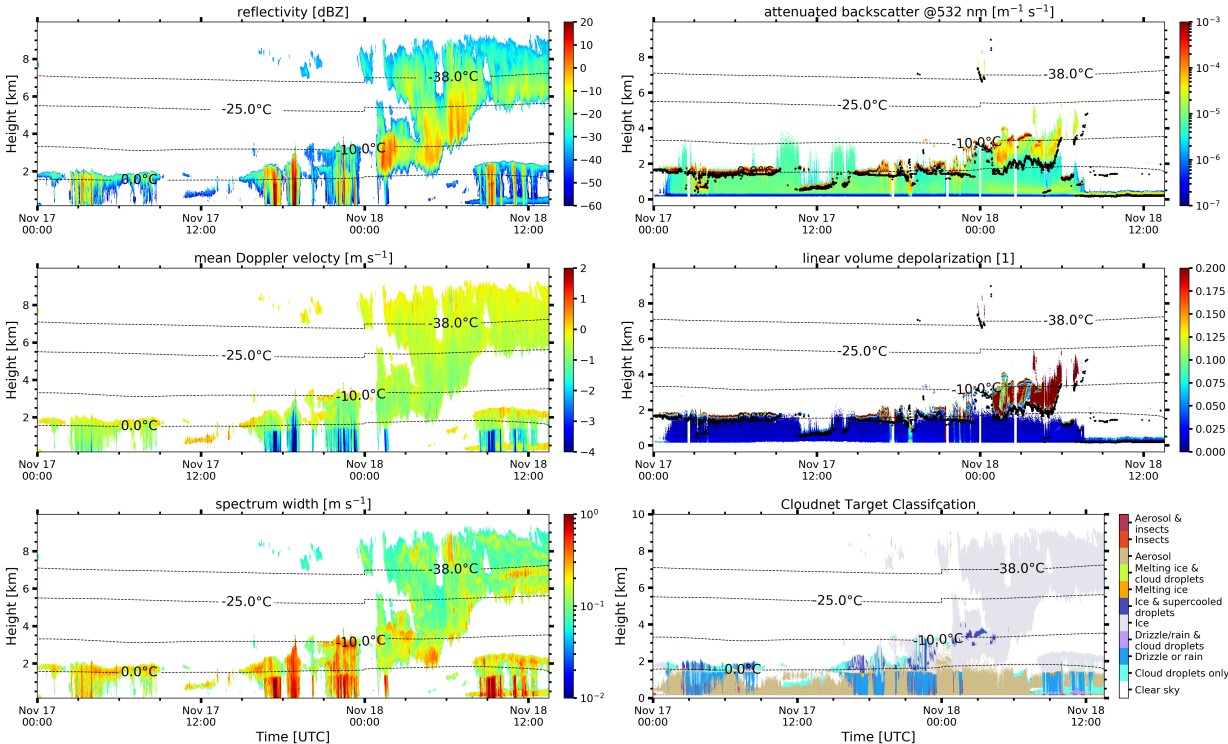

**Figure 1.** left: MIRA-35 radar reflectivity factor(top), radar mean Doppler velocity (middle), radar spectrum width (bottom), right: PollyXT lidar attenuated backscatter coefficient at @532 nm (top), PollyXT lidar linear volume depolarization (middle) and Cloudnet target classification (bottom) of Nov 17, 2014 00 UTC to Nov 18, 2014 13:30 UTC observed during the ACCEPT experiment in Cabauw, Netherlands. Black stars (in the lidar variable panels) indicate the first cloud base detected by the ceilometer.

## 2.2   Description of the used Artificial Neural Network

10   Luke et al. (2010) use collocated measurements with profiling cloud Doppler radar and polarization lidar in thin mixed-phase clouds or lower layers of thick mixed-phase clouds to provide information about the existence of liquid water in higher cloud



layers by predicting the lidar backscatter and depolarization signal from morphological features in the cloud radar Doppler spectrum. The procedure to determine the existence of supercooled-liquid droplets from cloud radar Doppler spectra is a two-step technique. In the first step, morphological feature extraction from cloud radar Doppler spectra is done by applying a second order Gaussian continuous wavelet transform (CWT) to each measured radar Doppler spectrum. In that way, the

spectral power is decomposed into a 2D-array with feature localization in Doppler velocity and spectrum width; each Doppler spectrum can thus be regarded as a sum of different Gaussians. In the second step, a selected subset of bins from six different scales of the CWT as well as the first three radar moments (reflectivity factor ($Z_e$ [dBZ]), mean Doppler velocity ($V_D$ [m s$^{-1}$]), and Doppler spectrum width ($\sigma$ [m s$^{-1}$])) of each Doppler spectrum are the input to the ANN used in this work to predict the existence of liquid. The ANN is of the multi-layer perceptron (MLP) type consisting of 256 input nodes, five hidden

layers, and two output nodes. Each of the five hidden layers consists of 32 nodes. Lidar particle backscatter coefficient ($\beta$ [sr$^{-1}$ m$^{-1}$] ) and lidar depolarization ratio ($\delta$) are the two output variables from which the existence of liquid is predicted using appropriate thresholds of $\beta$ and $\delta$ later on. In the training phase (which was performed on data from the Mixed Phase Arctic Clouds Experiment (MPACE, Verlinde et al. (2007)) obtained in fall 2004 at the U.S. Department of Energy's (DOE) Atmospheric Radiation Measurement (ARM) North Slope of Alaska (NSA) permanent site in Utqiagvik (formerly known

as Barrow), Alaska, the backpropagation of errors algorithm was applied. In short, the $\beta$ and $\delta$ output of the ANN for each time and height pixel were compared to values measured with a High Spectral Resolution Lidar (HSRL, Eloranta (2005)). The difference between ANN-predicted and lidar-observed (i.e., the error) was monitored and the internal weights of the nodes were adjusted until the error did not decrease any further during the successive cycling through the Doppler spectra training data set. Only a fraction of the MPACE data was considered in the training phase, most of the data was used for validation. Turbulent

broadening of the cloud radar Doppler spectrum (e.g. in strong convection) decreases the imprint of cloud microphysics on the Doppler spectra. The MPACE dataset was characterized by largely stratiform conditions. As stated in Gardner and Dorling (1998), the ability of an ANN to predict cloud properties does not only dependent on an informed choice of predictors but also requires sufficient data that fully represent all cases that the ANN is required to generalize, as ANNs perform well for interpolation but poorly for extrapolation. We can thus only expect good predictions of liquid in low-turbulent clouds but not

in strongly convective clouds. The objective of this study was to check the performance of the ANN trained with the MPACE observations in Luke et al. (2010) on a new data set, the ANN was thus not re-trained.

## 2.3 Classifying liquid containing sections from ANN-predictions

The ANN-predicts backscatter coefficient and particle depolarization ratio. Thresholds need to be applied to these predicted $\beta$ and $\delta$ in order to identify regions which show optical properties similar to the ones produced by liquid water.

For visual illustration of the mapping from predicted lidar variables to hydrometeor class labels, a scatter plot of predicted $\beta$ and $\delta$ was created (Fig. 2 (a)). As previously mentioned, lidar observed or ANN-predicted high values of $\beta$ and near-zero $\delta$ are reliable indicators of liquid-dominated cloud regions; they clearly stand out as a feature in Fig. 2 (a). The scatter plot of predicted $\beta$ and $\delta$ shows two more distinct features, one between the functions "linear-1" and "linear-2" with higher values of $\delta$ and lower values of $\beta$ indicating ice and another feature of very high values of $\delta$ and very low values of $\beta$ situated





below the function "linear-2" that can be attributed to the optically thinner ice cloud with lower radar reflectivities above 7 km on Nov 18, 2014 (see Fig. 2 (b)). Similar to Luke et al. (2010), fixed thresholds of $\beta$ and $\delta$ were used to derive a binary mask separating liquid predictions from other target types. For a sensitivity study of ANN-predicted liquid occurrence for the entire ACCEPT data set, different HSRL-based published thresholds (Shupe, 2007; de Boer et al., 2009; Luke et al., 2010) as well as a new linear function threshold (labeled "linear-1" in Fig. 2) were employed (see Table 1). Threshold values for $\beta$ of all three published studies are similar. Shupe (2007) and Luke et al. (2010) use the same $\delta$ threshold of 0.1 for liquid classification while de Boer et al. (2009) with a value of 0.03 is much more stringent. The studies are subsequently referred to as "Shupe2007", "deBoer2009", and "Luke2010". The linear-1 threshold function was found by a sensitivity study and gave the most similar classification results to the three cited published threshold values. Figure 2 (b) shows the corresponding time-height representation color coded by linear separation of the predicted (backscatter vs. depolarization) dimension using linear functions.

**Table 1.** Published thresholds of $\beta$ and $\delta$ for lidar-based liquid classification and linear-1 function threshold used for ACCEPT data set.

| method | thresholds |
|---|---|
| Shupe2007 | $\log(\beta) > -4.5, \delta < 0.1$ |
| deBoer2009 | $\log(\beta) > -4.3, \delta < 0.03$ |
| Luke2010 | $\log(\beta) > -4.3, \delta < 0.1$ |
| linear function-1 ($m\delta + \beta$) | $m = 12, \beta = -5.0$ |

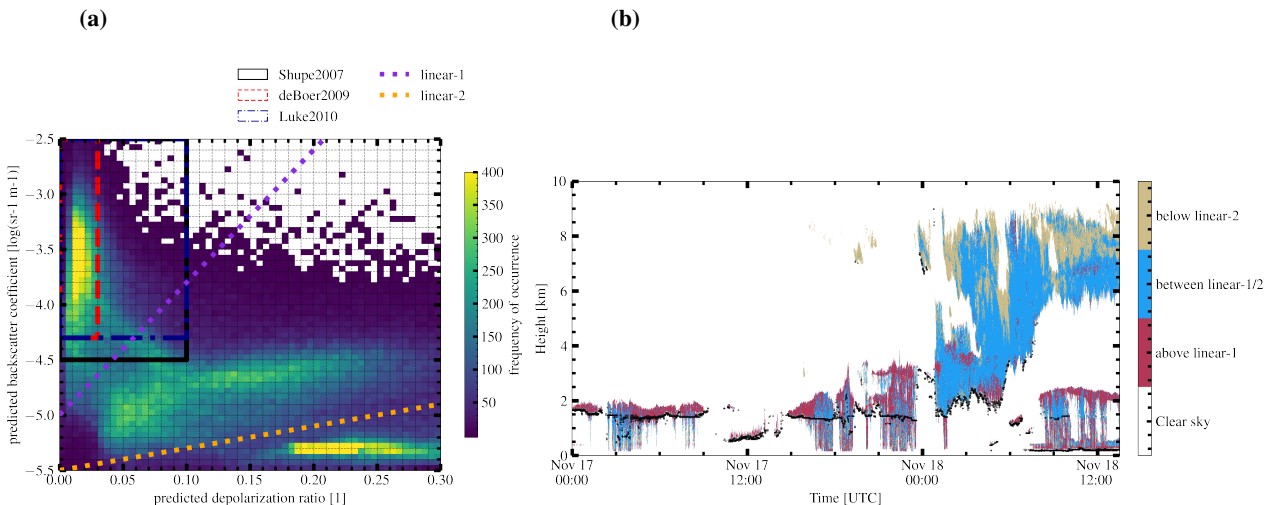

**Figure 2.** (a) Frequency of occurrence of ANN-predicted lidar backscatter coefficient $\beta$ vs. predicted lidar linear depolarization $\delta$ for the Nov 17 - 18, 2014 case study. (b) Time-height mapping of predicted $\beta$ and $\delta$ of the three corresponding areas in the (a) panel, which are separated by the two linear thresholds. Black dots indicate the ceilometer cloud base height.





The liquid classification methods were applied to the entire ACCEPT dataset. For doing so, the following pre- and postprocessing steps were applied to the seven-week long data set. Firstly, to account for the effects of radar partial beam filling, cloud edges are excluded from the ANN input data by setting data in the first and last range gate of a detected cloud (i.e. cloud base and cloud top pixel) to "clear sky". Secondly, only "cloud" pixels of the Cloudnet target classification mask (between first cloud

base and last cloud top) are considered, so that pixels classified as rain/drizzle and aerosols/insects were explicitly excluded. Thirdly, ANN liquid predictions for regions with good lidar echo and Cloudnet-classified as non-liquid class, are reclassified as non-liquid. Fourthly, using model temperature data of the Global Data Assimilation System (GDAS1) employed by the Global Forecast System (GFS) model, unphysical liquid predictions below $-37\,°C$ were re-classified as ice. The in-cloud pixels which were classified as liquid-containing by the ANN using the above-mentioned thresholds were sometimes quite patchy. Similar

to Shupe (2007) a homogenization step to create more coherent liquid layer structures, by using a 5x5 pixel neighborhood smoothing was introduced. A pixel was kept as liquid-containing pixel, when at least $60\,\%$ of the pixels in the 5x5 box around the center one were also classified as liquid-containing.

## 3   Results and Discussion

To assess the performance of the Luke et al. (2010) ANN-based liquid prediction from cloud radar Doppler spectra using different published thresholds of lidar backscatter coefficient and depolarization ratio against the Cloudnet target classification and against independent observables, a two-step validation was performed. Firstly, a case study (Nov 17-18, 2014 consisting of 100.000 samples) was analyzed in depth, see Table 2. Secondly, statistical results for the ANN-based liquid-prediction for the entire ACCEPT data set (1070 hours of observations, i.e. 1.7 million samples) are given in Table 3 and discussed subsequently.

In the following, the abbreviation CD is used for *cloud droplets bearing samples* and non-CD for *non-cloud droplets bearing samples*.

Predictions that meet the criteria from Section 2.3 are compared to classifications from Cloudnet (ground-truth). The comparison yields an error matrix consisting of correctly classified predictions, i.e. *true positive* (TP) and *true negatives* (TN) as well as *false positives* (FP) and *false negatives* (FN) which concerns wrong predictions, respectively. Described below are four

metrics used to evaluate the predictive performance against Cloudnets' liquid detection, three correlation coefficients $\rho_{a,b}$, and the fraction of liquid predicted located within a relative humidity above 90%.

– precision $= \frac{TP}{TP+FP}$: The precision value measures the amount of CD overestimation. The closer to 1, the fewer FP classification a method computes. (if precision $< 1 \Rightarrow$ CD overestimation)

– recall $= \frac{TP}{TP+FN}$: The recall value measures the amount of CD underestimation. The closer to 1, the fewer FN classifi-

cation a method computes. (if recall $< 1 \Rightarrow$ CD underestimation)

– accuracy $= \frac{TP+TN}{TP+TN+FP+FN}$: The closer the accuracy value gets to 1 the more pixel were classified correctly in an absolute and non-balanced way. (overall accuracy)





- F1 $-$ score $= \dfrac{2}{\frac{1}{\text{recall}} + \frac{1}{\text{precision}}}$ : The harmonic mean of CD over- and underestimation.

- $\rho_{\text{ceilo-CBH,LLH}}$: correlation of first ceilometer cloud base height (CBH) with first liquid layer height (LLH) of ANN and Cloudnet

- $\rho_{\text{MWR-LWP,LLT}}$: correlation of MWR-LWP with ANN- and Cloudnet-derived liquid layer thickness (LLT, product of sum of liquid containing pixel per profile times range gate resolution)

- $\rho_{\text{MWR-LWP,LWP}_{\text{ad}}}$: more physically meaningful correlation of MWR-LWP with LWP calculated from LLT and profiles of temperature and pressure under adiabatic assumption (as in Karstens et al. (1994))

- Liq-Pxl at RH $> 90\,\%$: fraction of liquid-classified pixels that overlap with pixels of radio-sounding based relative humidity (RH) with respect to water above 90 %

## 3.1 Case Study Results

The 37.5 h long case study of Nov 17, 2014 0 UTC - Nov 18, 2014 13:30 UTC was characterized by a multitude of cloud types including pure liquid water clouds, stratiform mixed-phase clouds, high clouds, mid-level clouds and near-surface clouds (fog) as shown in Fig. 1. On Nov 17, 2014 between 3-9 UTC and 15-0 UTC several rain showers from low mixed-phase clouds with cloud-top temperatures between $-10\,°C$ and $-2\,°C$ were observed. At around 12 UTC, a thin warm liquid cloud at 1 km altitude with a LWP below $30\,\text{g m}^{-2}$ was present. On Nov 18, different multi-layer clouds with varying vertical extent were present, a high cloud in 6-9 km was firstly situated above a mid-level cloud in 2-5 km and later on over a precipitating stratiform cloud in about 2 km altitude with cloud-top temperature of $-5\,°C$ which had a layer of near-surface fog below.

In Fig. 3 the resulting liquid masks of all presented thresholds and Cloudnet for this case study are shown. There is mostly good agreement in liquid-detection for the stratiform mixed-phase cloud and the liquid cloud in 1 km on Nov 17. However, since the liquid-threshold boundaries of deBoer2009 are very strict, many potential liquid pixel candidates are not considered (e.g. around 3 UTC, and 18 UTC on Nov 17). For this particular case, the Cloudnet algorithm was not able to fully identify the cloud-top layer at $-10\,°C$ during 0-6 UTC and at 2 km during 9-12 UTC on Nov 18, as mixed-phase and/or supercooled liquid because of full lidar signal attenuation in the rain/fog below. The ANN-based liquid-detection outperforms Cloudnet in these situations. However, as stated in Luke2010, the ANN performance is expected to decrease in more turbulent conditions leading to Doppler spectrum broadening. This is the case at 10-13 UTC on Nov 19 in a layer at about 7 km altitude around $-37\,°C$ (very low probability for liquid) for all but the deBoer2009 threshold.

For independent validation of the areas classified as liquid-containing, the summed up liquid layer thickness (LLT) of all pixels classified as liquid by the ANN or Cloudnet is compared to the MWR-LWP (Figure 4) as proposed by Luke2010. Profiles in which rain/drizzle reached the ground were excluded in the LLT-determination to avoid situations with a wet MWR radome leading to an invalid MWR-LWP estimate (as indicated by the rain flag in Figure 4). In some situations the ANN and in others Cloudnet matches the timeseries of MWR-LWP better on Nov 17 while on Nov 18 the ANN-LLT mostly (except



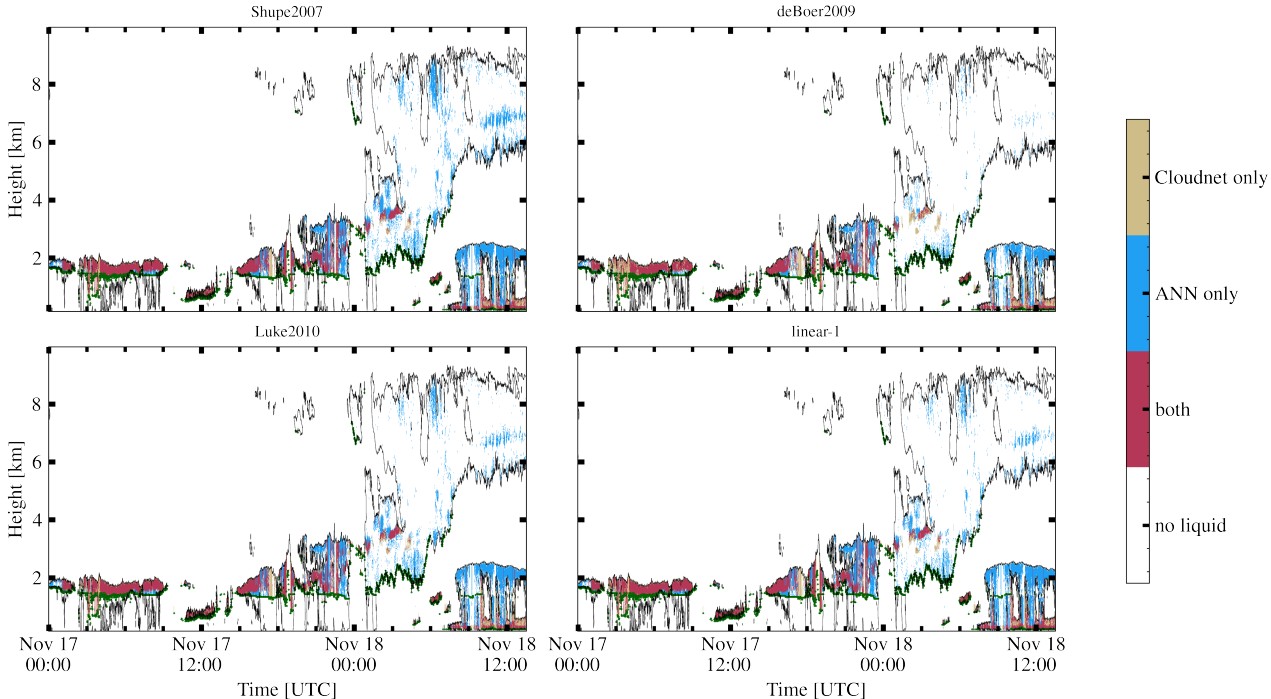

**Figure 3.** Sensitivity study of liquid pixel classifications of the Nov 17 - 18, 2014 case study using liquid mask thresholds of Shupe2007 (upper left), deBoer2009 (upper right), Luke2010 (lower left), linear-1 (lower right) on the ANN-predicted lidar variables. Light-brown: Cloudnet only liquid classifications, blue: ANN-predicted pixels using the given thresholds which were not classified as liquid by Cloudnet, red: pixels classified as liquid by the ANN and Cloudnet, black contour: cloud edges, green circles: ceilometer first cloud base height, white: clear-sky or not classified as liquid by either method.

when using the deBoer threshold) overestimates LLT due to the explained misclassifications. More ACCEPT case studies have been analyzed in detail but are not presented here because they show very similar results.

The error matrix and evaluation metrics (first 8 rows in Table 2) show the performance of Luke et al. (2010) by comparing the predictions to valid Cloudnet liquid detections. Depending on the threshold given in Table 1, precision ranges between 0.9
5   (Shupe2007) and 0.92 (deBoer2009). Contrarily, recall values range between 0.53 (deBoer2009) and 0.67 (linear-1) indicating that more lose thresholds are better in detecting more TP, while keeping the number of FN comparably low. Overall accuracy ranges between 0.78 (deBoer2009) and 0.83 (linear-1), with the more balanced F1-score showing the same threshold candidates for minimum 0.67 (deBoer2009) and maximum 0.77 (linear-1) values. An F1-score close to one means low number of FP and low number of FN, correctly identifying real CD ,while not being disturbed by false alarms. Regions with high Doppler
10   spectrum width at cloud base (see Figure 1 Nov. 18, 3-6 UTC between 2 and 3 km altitude) contribute to a large portion of those FP for all thresholds. Lower recall values indicate a higher degree of underestimation of CD detections, which is caused



by liquid layers with low LWP values below $50\,\mathrm{g\,m^{-2}}$, e.g. the thin liquid cloud on Nov. 17 around 12 UTC in 0.5 to 1 km altitude.

In this work the ceilometer first cloud base height variable is correlated to the predicted first liquid layer height (if liquid is present). However, in some situations, like on Nov 18, 2014 between 1–4 UTC, the ceilometer cloud base variable is not

representing the base of the liquid layer but instead the base of precipitating ice crystals (Fig. 1). Situations like these are the most important factor for leading to $\rho_{\mathrm{ceilo\text{-}CBH,LLH}}$ of the four ANN methods to be on the order of 0.86 (deBoer2009) to 0.92 (Shupe2007), i.e., a failure rate of 8–14 %. $\rho_{\mathrm{ceilo\text{-}CBH,LLH}}$ for ceilometer-CBH and Cloudnet (which uses ceilometer data as input for the thermodynamic phase classification) are expected because the ceilometer is better able to detect thin liquid water layers with low LWP than the cloud radar is. This being said, $\rho_{\mathrm{ceilo\text{-}CBH,LLH}}$ for Cloudnet is actually expected to be 100 %- it is

slightly lower due to averaging of ceilometer data to the 30 s Cloudnet input file resolution.

The $\rho_{\mathrm{MWR\text{-}LWP,LLT}}$ also shows positive correlations for all methods. It ranges between 0.44 (Shupe2007) to 0.53 (deBoer2009), for Cloudnet the $\rho_{\mathrm{MWR\text{-}LWP,LLT}}$ amounted to 0.47. Converting the LLT to the physical more meaningful $\mathrm{LWP_{ad,cor}}$ results in $\rho_{\mathrm{MWR\text{-}LWP,LWP_{ad}}}$ that are very similar to $\rho_{\mathrm{MWR\text{-}LWP,LLT}}$ with moderate correlation (0.47) for deBoer2009, and weaker correlations for all other methods. Both $\rho_{\mathrm{MWR\text{-}LWP,LLT}}$ and $\rho_{\mathrm{MWR\text{-}LWP,LWP_{ad}}}$ of deBoer2009 show the strongest relationship to the measured

MWR-LWP. The period Nov. 17, 21 to Nov. 18, 12 UTC in Figure 4 shows major the differences in LLT between the de-Boer2009, Cloudnet and the other methods. The number of CD predictions in the precipitating system (Nov. 17, 20-23 UTC), the region with higher spectrum width (Nov. 18, 4-6 UTC at cloud base and Nov. 18, 10-13 UTC at 7 km altitude), and turbulence at cloud top (Nov. 18, 6-7 UTC at 8 km) (see: Fig. 3) are lowest for deBoer2009, therefore reflecting the MWR-LWP best. However, deBoer2009 also counts the least amount of TP, due to its tight thresholds, which seems to have minor effects

on the correlation coefficient. Unfortunately, no radio sondes were launched during the presented case study, so the relative humidity related measure could not be determined. Multiple other cases studies where conducted resulting in similar results.

### 3.2 Statistical results for entire ACCEPT field campaign

A second, more general evaluation of all methods is done for the entire ACCEPT field campaign comprised of 1070 h of observations counting more than 1.7 M samples. The summary of this evaluation is presented in Table 3. All thresholds achieve

high precision values $> 0.9$, indicating a low FP rate. Recall values are moderately lower compared to the presented case study, ranging from 0.4 (deBoer2009) to 0.54 (Shupe2007). Accuracy lies above 0.75 (three out of four predictions are correct) for all methods except slightly lower values for deBoer2009 (explained in Section3.1). The F1-score range between (0.56-0.67) , again with the lowest value achieved by deBoer2009. However, deBoer2009 achieves best correlation for $\rho_{\mathrm{MWR\text{-}LWP,LLT}}$ and $\rho_{\mathrm{MWR\text{-}LWP,LWP_{ad}}}$, due to CD overestimation (larger numbers of FP ) for Shupe2007, Luke2010 and linear-1. Overall, all methods

achieve better correlation values for the entire data set compared to the case study, with high $\rho_{\mathrm{ceilo\text{-}CBH,LLH}}$, values ranging from (0.86-0.92) and (0.97) for Cloudnet respectively.

The values in Table 2 and Table 3 indicate that the entire data set is well represented by the chosen case study. As indicated in Section 3.1, missing spectral signatures and turbulence broadened radar Doppler spectra are the main driver for miss-classifications of the pre-trained Luke et al. (2010) approach. An additional independent validation is done using radiosonde





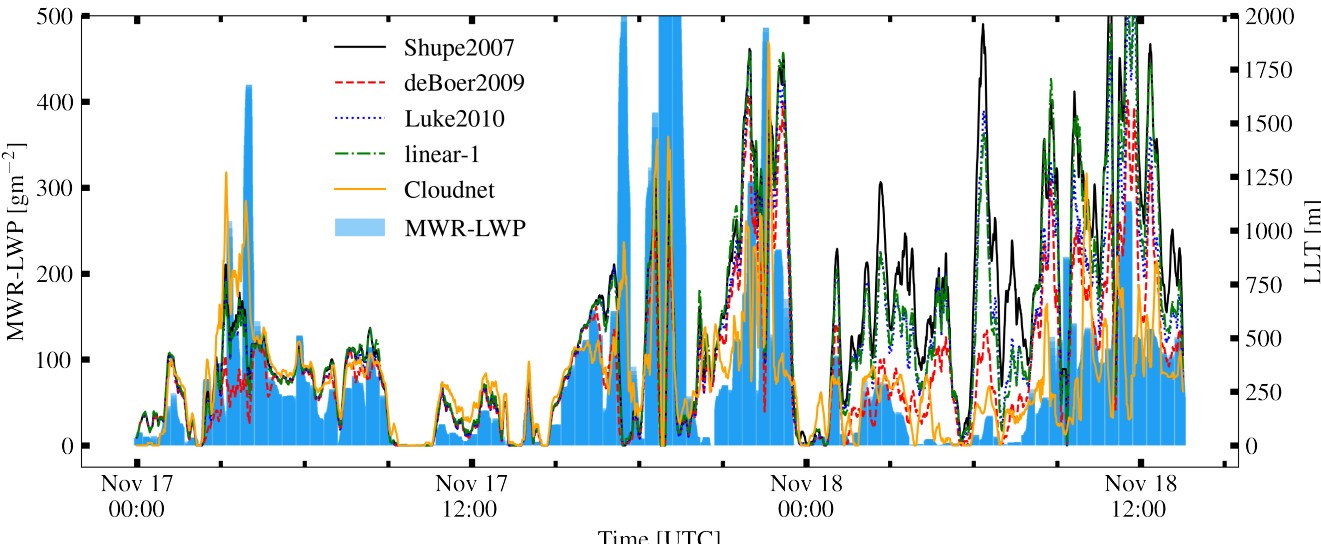

**Figure 4.** Comparison of MWR-LWP (left y-axis, blue bars) and liquid layer thickness (LLT, right axis) of the ANN-predicted liquid layer mask (red) and Cloudnet-LLT (black) for the Nov 17 - 18, 2014 case study for all used liquid-detection thresholds. Green and red dots near the bottom of the plots respectively indicate valid MWR-LWP data (no drizzle or rain detected + 15 min for the MWR radome to dry), and invalid MWR-LWP data masked for analysis due to precipitation. (Garrett and Peng, 2021)

**Table 2.** Error matrix, performance metrics, and correlation coefficients for ceilometer-CBH vs. LLH, MWR-LWP vs. LLT, MWR-LWP vs. $LWP_{ad,cor}$, case study Nov 17-18. Statistic includes only valid pixel.

|  | Shupe2007 | deBoer2009 | Luke2010 | linear-1 | Cloudnet |
|---|---|---|---|---|---|
| TP | 28684 | 22209 | 26803 | 28816 | 46215 |
| TN | 59342 | 60605 | 59615 | 59620 | 62424 |
| FP | 3082 | 1819 | 2809 | 2804 | 0 |
| FN | 17531 | 24006 | 19412 | 17399 | 0 |
| precision | 0.903 | 0.924 | 0.905 | 0.911 | 1 |
| recall | 0.621 | 0.481 | 0.580 | 0.624 | 1 |
| accuracy | 0.810 | 0.762 | 0.795 | 0.814 | 1 |
| F1-score | 0.736 | 0.632 | 0.707 | 0.740 | 1 |
| $\rho_{MWR\text{-}LWP,LLT}$ | 0.436 | 0.533 | 0.490 | 0.489 | 0.471 |
| $\rho_{MWR\text{-}LWP,LWP_{ad}}$ | 0.275 | 0.471 | 0.335 | 0.345 | 0.399 |
| $\rho_{ceilo\text{-}CBH,LLH}$ | 0.775 | 0.725 | 0.738 | 0.755 | 0.913 |
| Liq-Pxl at RH > 90 % | n/a | n/a | n/a | n/a | n/a |





**Table 3.** Error matrix, performance metrics, and correlation coefficients for ceilometer-CBH vs. LLH, MWR-LWP vs. LLT, MWR-LWP vs. $\text{LWP}_{\text{ad,cor}}$, for the entire ACCEPT data set. Statistic includes only valid pixel.

|  | Shupe2007 | deBoer2009 | Luke2010 | linear-1 | Cloudnet |
|---|---|---|---|---|---|
| TP | 406235 | 302643 | 374880 | 401331 | 757342 |
| TN | 919571 | 938429 | 925740 | 925243 | 962586 |
| FP | 43015 | 24157 | 36846 | 37343 | 0 |
| FN | 351107 | 454699 | 382462 | 356011 | 0 |
| precision | 0.904 | 0.926 | 0.911 | 0.915 | 1 |
| recall | 0.536 | 0.400 | 0.495 | 0.530 | 1 |
| accuracy | 0.771 | 0.722 | 0.756 | 0.771 | 1 |
| F1-score | 0.673 | 0.558 | 0.641 | 0.671 | 1 |
| $\rho_{\text{MWR-LWP,LLT}}$ | 0.490 | 0.566 | 0.515 | 0.530 | 0.473 |
| $\rho_{\text{MWR-LWP,LWP}_{\text{ad}}}$ | 0.348 | 0.462 | 0.370 | 0.387 | 0.432 |
| $\rho_{\text{ceilo-CBH,LLH}}$ | 0.915 | 0.859 | 0.897 | 0.905 | 0.974 |
| Liq-Pxl at RH > 90 % | 0.602 | 0.653 | 0.626 | 0.620 | 0.816 |

launches from the campaign site as well as launches from DeBilt airport about 30 km away. Liquid-detected pixels are only evaluated in this way within ±30 min of a radiosonde launch, meaning only a small subset of data from the entire field experiment is considered. Radio sounding profiles with RH with respect to liquid water (w.r.t.l.) larger than 90 % overlapping with liquid detection layers occur only during 1.5 h out of 58 h of available liquid detection data, i.e. only during 2.5 % of

5 the time is liquid classified. This validation method thus only has limited utility for the quality of the thermodynamic phase classifications made, but is shown here for the sake of completeness as similar future evaluation studies might have larger datasets available. However, for all methods the majority of number of liquid-containing pixels occur when the radiosonde RH w.r.t.l. is larger than 90 % and liquid occurrence is thus likely. There are two explanations why the fraction of Cloudnet-classified liquid pixel overlapping with areas of radiosonde RH > 90 % is much higher (72 %) than for the ANN results (54-

10 61 %). Firstly, with the radiosonde drifting away with height (and time), the assumption of having the same thermodynamic profile over the ACCEPT-site and the sounding location becomes less certain for liquid detections higher in the atmospheric profile (where the ANN is predicting more liquid than Cloudnet). Secondly, the overlap fraction does include false positives (mostly caused by the ANN) but not true negatives.





## 4 Summary and Outlook

The current study shows that synergistic observations of depolarization lidar and cloud Doppler radar in conjunction with machine learning techniques can be used to detect liquid beyond full lidar signal attenuation. It was shown that this approach performs well in stratiform (low-turbulent) cloud situations but is not suited for strongly convective situations in which the imprint of different hydrometeor populations in the same cloud volume on the cloud radar Doppler spectrum is masked by turbulent spectrum broadening. We demonstrated that the ANN of Luke et al. (2010) pre-trained with the MPACE data set in Alaska could successfully be applied to the ACCEPT data set obtained in Cabauw, the Netherlands and is able to improve the Cloudnet target classification for stratiform optically thick liquid-layers or situations in which multiple liquid layers exist. We applied different published lidar-based liquid-detection thresholds to the predicted lidar backscatter coefficients and de-polarization lidars - all were found to perform better in some situations than others and could be seen as either to stringent (deBoer2009) missing thinner liquid layers or to broad (Shupe2007, Luke2010, "linear-1") leading to misclassification of ice as liquid. No suggestion on best thresholds can thus be made. To overcome limitations due to ambiguities caused by thresholding, focus should therefore be put on the development of techniques which do not rely on explicit lidar thresholds for liquid detection. This could be realizable by applying novel convolutional artificial neural networks which could be used to exploit the full information content of high-resolution cloud radar Doppler spectra.

The identification of the presence of liquid layers in the entire vertical column of optically thick or multi-layered cloud situations is a first step to get a better understanding of which microphysical particle growth processes might occur in a mixed-phase cloud. The shown results will therefore be used in follow-up studies for characterization of microphysical hydrometeor growth processes. Our study also shows that it might be possible in future to identify the suitability of a cloud region for the identification of liquid water based on cloud radar Doppler spectra analysis. One approach could be the application of a stratiform/convection separation algorithm.

*Data availability.* Cloudnet-processed data for the ACCEPT campaign are available via https://cloudnet.fmi.fi. The Mira-35 moment data as well as compressed (noise-removed) Doppler spectra are available upon request from Patric Seifert (seifert@tropos.de).

*Author contributions.* Heike Kalesse and Willi Schimmel, did the data analysis and prepared the manuscript. Patric Seifert was PI of the ACCEPT field experiment and helped in preparing the manuscript. Edward Luke did the ANN simulations.

*Competing interests.* No competing interests are present.





*Acknowledgements.* Most of the work H. Kalesse-Los conducted for this study was within the framework of the DFG project COMPoSE, GZ: KA 4162/1-1. Willi Schimmel was funded through the ESF-Project 100339509. Das Vorhaben "Bodengebundene Fernerkundung der Atmosphäre zur Verbesserung der Charakterisierung mikrophysikalischer Wolkeneigenschaften sowie der Leistungsprognose erneuerbarer Energien FKZ: 100339509" wird im Rahmen des Programms "Vorhaben in den Bereichen Hochschule und Forschung" vom Freistaat Sachsen und dem Europäischen Sozialfonds gefördert. This project has also received funding from the European Union's Horizon 2020 research and innovation programme under grant agreement No 654109.



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
