# Peer review of "Evaluating cloud liquid detection against Cloudnet using cloud radar Doppler spectra in a pre-trained artificial neural network"

_Atmospheric Measurement Techniques, 2021_

## Referee Comment (RC1)

Review: Evaluating cloud liquid detection using cloud radar Doppler spectra in a pre-trained artificial neural network against Cloudnet liquid detection
By Authors: Heike Kalesse, Willi Schimmel, Edward Luke, and Patric Seifert

**General comments:**

This study compares the performance of different techniques (#1 lidar (Cloudnet product); #2 ANN applied to radar Doppler spectra) in detecting supercooled liquid water. The ANN method (Luke et al., 2010) trained with radar Doppler spectra data obtained in the Arctic was applied to radar data recorded during the ACCEPT in Cabauw, the Netherlands, 2014. The comparison was firstly conducted to an event with some analysis, and then statistical results for the whole campaign were presented. The results show that the ANN approach outperforms the Cloudnet algorithm in multi-layered stratiform clouds. This is actually expected since Cloudnet target classification is relied on lidar data which have been totally attenuated by the lowest liquid layer. I feel the most interesting part of the study will be assessing the performance of ANN method in presence of convection. I will list more specific comments below.

Overall, this manuscript is well-written and relevant studies are properly cited. I like the Introduction section, since it is quite comprehensive and reads very friendly for researchers who are not very familiar with this topic. Some parts in results section need further clarification. The scientific merit that just showing the (Luke et al., 2010) works well in a different climatology seems weak to me. I think the evaluation in presence of convection could be supplemented by more detailed analysis. Then, this could be a nice paper that I would recommend to be published on AMT.

**Major comments:**

1) It appears that the method works well as shown by rho_ceilo-CBH, LLH > 85% in Table 2. However, I think it is more important to know at what conditions the 15% fails. Although it is already well known that the liquid peak in Doppler spectrum can be blurred by turbulence, at what extent the turbulence can smear the liquid peak is still not very clear. This may be described by factors related to turbulence, such as spectrum width, velocity, Z, variance of V and so forth. Then, the scientific significance of this study will be improved.

I think the current explanation is widely accepted knowledge. As the author wrote 'the objective of this study was to check the performance of the ANN trained with the MPACE observations in Luke et al. (2010) on **a new data set**'. The clouds over the Netherlands are definitely more convective than the Arctic, therefore the convective conditions should be well addressed.

Also, one explanation for the FP of ANN is enhanced SW. To my understanding, the enhanced SW should smear the liquid signature, thus leads to FN. So, turbulence can lead to FP and FN. At what conditions those two 'bad' classes can be expected?

2) Figure 2a. I am curious how well the ANN can predict $\beta$ and $\delta$. This may also be a part of the

'evaluation'. The accuracy of estimated β and δ may be as important as the selection of thresholds as presented in Table 1. Have you compared the predicted values with observations? At least with beta observed by the ceilometer.

3) P5 L6.' Thirdly, ANN liquid predictions for regions with good lidar echo and Cloudnet-classified as non-liquid class, are reclassified as non-liquid.' This step confuses me. I think it is of importance to know at what conditions the ANN misclassifies lidar-detected non-liquid to liquid. I would not simply ignore this scenario.

**Minor commennts:**

Figure 1. Numbers for subfigures are missing.

P2 L23 D^6 comes from Rayleigh approximation, which may not be valid for a large fraction of large ice crystals for a cloud radar

Figure 2 and 3. I suggest overlap the temperature isothermal lines which will greatly help the interpretation of the results.

Figure 3. The green circles are hardly identifiable from black cloud edges. Please use the color which is more contrasting with others.

Figure 3. The overview of this precipitation event has already been presented on Figure 2 (b). I suggest the use of smaller yaxis range. Most interesting signatures are below 2 km. The current yaxis scale seems too large to me and the differences among these subfigures are difficult to recognize. The liquid layer above 4 km may deserve a separate figure.

Figure 3. (Although I doubt the reasonability of 'Thirdly, ANN liquid predictions for regions with good lidar echo and Cloudnet-classified as non-liquid class, are reclassified as non-liquid.') The region marked by red circle should correspond to 'good lidar echo' in Figure 1. Why ANN still identified liquid in this region?

[Figure]

P9 L22. 'cloud-top layer at −10 °C during 0-6 UTC'. I am confused by this sentence. -10 °C is around 3.5 km. The cloud top during 0-6 UTC Nov 18 is definitely much higher. Do you mean 21-24 UTC Nov 17?

P9 L25. This is interesting. Turbulence favors liquid formation, but may lead to weakened liquid spectral signature if liquid is present. As shown in Figure 1, it is obvious that the SW is enhanced at this layer. However, given the weak signal in deBoer2009 and the rather low temperature, it is very unlikely that they are liquid layers. Could you please present examples of the radar Doppler spectrum in this layer as well as at 8 km 6 UTC Nov 18?

Figure 4. The rain flag is missing.

P10 L1. May not be the 'Misclassification'. In some cases, e.g. after 7 UTC Nov 18, lidar signals are totally attenuated by the lowest liquid. So, ANN may be correct in the upper layer. Please carefully address this point.

P10 L4. 'by comparing the predictions to valid Cloudnet liquid detections'. Do you mean the cloudnet product with 'good lidar echo'? Or regardless of the lidar signal quality?

P11 L7. To my understanding, high $\rho$ ceilo-CBH,LLH for ceilometer-CBH and Cloudnet is expected, because cloudnet uses ceilometer data as input. How is this linked to the sensitivity between lidar and radar? I am confused by the logic.

P11 L9. How the averaging affects the performance?

P13 L10. The first point may explain the difference between radiosonde and cloud/lidar method, but not the reason why the liquid pixel is higher in cloudnet than ANN.

P14 L15. It would be nice to refer the relevant machine learning techniques. For example, the work by the authors (Kalesse et al., AMT, 2019).

Typos:
P9 L25. Nov 18

P11 L7. The high value of rho_ ceilo-CBH,LLH … is expected, because…

P11 L21. Case; resulted

---

## Author Comment (AC2)

reply to reviews of "amt-2021-60"
https://amt.copernicus.org/preprints/amt-2021-60/amt-2021-60.pdf
Title: Evaluating cloud liquid detection using cloud radar Doppler spectra in a pre-trained artificial neural network against Cloudnet liquid detection
Author(s): Heike Kalesse-Los et al.
MS No.: amt-2021-60
MS type: Research article

We thank both reviewers for their comments which we addressed in a point by point way below and which resulted in major additions to the manuscript. Reviewer comments are in black, our replies in green.

Reviewer #1

**General comments:**

This study compares the performance of different techniques (#1lidar (Cloudnet product); #2 ANN applied to radar Doppler spectra) in detecting supercooled liquid water. The ANN method (Luke et al., 2010) trained with radar Doppler spectra data obtained in the Arctic was applied to radar data recorded during the ACCEPT in Cabauw, the Netherlands, 2014.The comparison was firstly conducted to an event with some analysis, and then statistical results for the whole campaign were presented.The results show that the ANN approach outperforms the Cloudnet algorithm in multi-layered stratiform clouds.This is actually expected since Cloudnet target classifications relied on lidar data which have been totally attenuated by the lowest liquid layer. I feel the most interesting part of the study will be assessing the performance of ANN method in presence of convection. I will list more specific comments below.
Overall, this manuscript is well-written and relevant studies are properly cited. I like the Introduction section, since it is quite comprehensive and reads very friendly for researchers who are not very familiar with this topic.Some parts in results section need further clarification.The scientific merit that just showing the (Luke et al., 2010) works well in a different climatology seems weak to me. I think the evaluation in presence of convection could be supplemented by more detailed analysis. Then, this could be a nice paper that I would recommend to be published on AMT.

General reply:
As stated by the reviewer, we show that the ANN approach outperforms the Cloudnet liquid classification in multi-layered clouds. While this indeed is not surprising, we believe that there is a need to make the Cloudnet community more sensitive to the limitations of the built-in lidar-only based Cloudnet liquid classification. This is for example corroborated by numerous papers giving Cloudnet-based cloud phase statistics (e.g. Fig 8 and Fig 9 of https://acp.copernicus.org/articles/19/4105/2019/, of Fig 11 of https://acp.copernicus.org/articles/21/289/2021/, etc), or Figure 16 of

https://rmets.onlinelibrary.wiley.com/doi/full/10.1002/qj.3971 , where the underestimation of observed w.r.t. modelled liquid is increasing with height (likely because of the decreasing likelihood that lidar can detect liquid water).
Our objective of this manuscript is to show that there are methods that do better in detecting liquid than the current Cloudnet algorithm in certain conditions. We will describe more state-of-the-art ANN and cloud climatologies in subsequent studies (in progress) and thus refrain from expanding the current manuscript too much.

**Major comments:**

1) It appears that the method works well as shown by rho_ceilo-CBH,LLH > 85% in Table 2. However, I think it is more important to know under what conditions the 15% fails.

True, trying to assess under which conditions the liquid-layer height (LLH) of ANN and ceilometer cloud base height (CBH) differ is important. We thus expanded the section on rho_ceilo-CBH,LLH accordingly to address the failure rate more in depth.:
"In this work the ceilometer first cloud base height variable is correlated to the predicted first liquid layer height (if liquid is present). rho_ceilo-CBH,LLH of the four ANN methods are on the order of  0.86 (deBoer2009) to 0.92 (Shupe2007) for the entire ACCEPT dataset, i.e., there is a failure rate of 8--14%. This failure rate can be explained by several conditions: Firstly, in some situations, like on Nov 18, 2014 between 1-4 UTC, the ceilometer cloud base variable is not representing the base of the liquid layer but instead the base of precipitating ice crystals. This is caused by specular reflection from the planar planes of horizontally aligned ice crystals as described in Westbrook  et al., 2010. As shown in the comparison of ceilometer CBH and ANN LLH below, when the ANN is not sensitive to these ice crystals, the difference in ceilo-CBH and ANN-LLH is high. Secondly, there are situations where liquid layers with low LWP are only detected by the ceilometer but not by the cloud radar (Nov 17, 11 UTC, 1.7 km) and thirdly, there are cloud scenes where the ceilometer is fully attenuated by precipitation or low level fog (thus reporting the precipitation base or fog base as first cloud base, see Figure below) which the radar can penetrate/is not sensitive to or which is below the first radar range gate. Fourthly, in situations where the ceilometer is still able to penetrate light precipitation to detect CBH (Nov 17, 3-9 UTC, 17-24 UTC) and the ANN misclassifies drizzle/rain as cloud droplets, further discrepancies arise. These conditions lead to a decrease of the rho_ceilo-CBH,LLH. The rho_ceilo-CBH,LLH for ceilometer-CBH and Cloudnet for the entire ACCEPT data set is higher and amounts to 0.97. While the cloud base height variable in Cloudnet is based on the gradient of ceilometer attenuated backscatter coefficient, the internal ceilometer cloud base determination is not precisely documented in the ceilometer manual. Differences in cloud base height leading to a failure rate of 3% may thus occur due to different backscatter coefficient thresholds."

As an example, we show in Fig. 1 below a comparison of ceilometer first cloud base height (CBH) and first liquid layer base height (LLH) from the ANN and Cloudnet below, which illustrates the larger differences between ceilometer and ANN leading to lower rho_ceilo-CBH,LLH.

[Figure]

Fig 1: Top: Time series of difference of ceilometer first cloud base height variable  and first liquid layer base from ANN (red) and Cloudnet (black) for Nov 17-18 2014 case study.
Bottom: Time series of ceilometer first cloud base height (CBH) variable (green) and first liquid layer base height (LLH) from ANN (red) and Cloudnet (blue) for Nov 17-18 2014 case study.

Rev1: Although it is already well known that the liquid peak in Doppler spectrum can be blurred by turbulence, at what extent the turbulence can smear the liquid peak is still not very clear. This may be described by factors related to turbulence, such as spectrum width, velocity, Z, variance of V and so forth. Then, the scientific significance of this study will be improved. I think the current explanation is widely accepted knowledge. As the author wrote 'the objective of this study was to check the performance of the ANN trained with the MPACE observations in Luke et al. (2010) on **a new data set**'. The clouds over the Netherlands are definitely more convective than the Arctic, therefore the convective conditions should be well addressed.

Thank you for this remark. We considered this problem by trying to assess conditions in which the ANN algorithm does well in predicting liquid or not by doing the following:
    a) analysis of radar moments for error matrix elements TP, FP, TN, FN in terms of Frequency of Occurrence (FoO) plots and 3D scatter plots as well as of time-height masks of error matrix elements
    b) determination of convective index kappa
which we are elaborating on subsequently.

    a) To evaluate the performance of the ANN for liquid detection, we created normalized frequency of occurrence (FoO) histograms for the entire ACCEPT data set (Fig 2) to see in which way the elements of the ANN cloud droplet prediction error matrix (TP, TN, FP, FN) differ in terms of radar moments (radar reflectivity factor Ze, mean Doppler velocity Vd, spectrum width, linear depolarization radio LDR) and environmental temperature.

What is evident in Fig2 below, is that the distribution of radar moments of TP is different from TN, FP, and FN while the FoO of radar moments of the latter TN, FP, FN are mostly similar. Specifically,

- Ze: the FoO of reflectivities of TP of liquid droplets is monomodal with a maximum occurrence at -25dbZ to -35dBZ, it is bimodal for TN, FP, FN with the two maxima occurring at -25dBZ and -10dBZ
- Vd: With values between -2 m/s and + 0.5 m/s the distribution of Vd of TP is narrower than of TN, FP, FN which have values of about -4 m/s and + 1 m/s and a maximum FoO at more negative values of around -0.5 m/s than the TP (max. FoO at -0.2 m/s)
- spectrum width: TP generally occur at larger spectrum width than TN, FP, FN with a max. FoO of TP at 0.2-0.25 m/s while the max. FoO of TN, FP, FN peaks at 0.05-0.1 m/s
- LDR: cloud droplets are spherical, their theoretical radar linear depolarization ratio (LDR) is thus minus infinity dB, due to technical limitations, the smallest LDR of the MIRA cloud radar is -30 dB. The FoO of TP exhibits a narrow peak around -30 dB and then declines sharply to low FoO values of higher LDR. Again, LDR FoO of TN, FP, FN are more similar to each other than to TP. However, while the max. FoO of FN is also at - 30 dB, the max. FoO of TN and FP spans a broad range between -30 dB and -24 dB. The FoO of increasingly higher LDR values decreases rapidly with an especially sharp decrease at -20 dB for all elements of the error matrix.
- environmental T: Plausibly, at lower temperatures T, more TN than TP occur. Most TP were found at T > 0 C which owes to the consecutive extinction of ground-based lidar signal with height, where the atmospheric T decreases.
- Comparing FoO of liquid detection error matrix with respect to the different thresholds, the more stringent criteria of deBoer2009 generally lead to narrower FoO distributions.

→ We included the FoO plot below and part of these explanations in Section 3 of the manuscript where the ANN-liquid prediction scores of the entire ACCEPT field experiment are discussed (please see "diff-pdf file").

[Figure]

Fig2: Normalized frequency of occurrence (FoO) of radar moments (Ze, Vd, spectrum width, LDR) and temperature of error matrix elements (TP, TN, FP, FN) of ANN-based liquid detection for the studies employing different thresholds of lidar backscatter and depolarization for the entire ACCEPT field experiment.

Another way to compare the error matrices of the ANN-predicted liquid occurrences is via 2D or 3D scatter plots of the first three radar moments Ze, Vd, and spectrum width for the entire data set. A complete view of the 3D scatter plot are shown in the following youtube videos: https://www.youtube.com/watch?v=Z9Z4ui8f4Z0 and https://www.youtube.com/watch?v=gz5iL51EzZw with (black=TP, red=FN, gold=FP, green=TN). It is evident that the error matrix elements mostly overlap and do not show distinctive separate clusters, meaning that the same combination of Ze, Vd, width can result in TP, TN, FP, FN. Same is true for the 2D scatter plots (not shown).
However, a few features in the 3D scatter plot are noteworthy:

- TP generally have higher Doppler spectrum width over the entire covered Ze values (as in FoO Fig 2 of this reply)
- FN and TN both can occur at high Ze. For Ze > 20 dBZ, the FN have large negative Vd and spectrum width < 0.6 m/s (rain/drizzle, see time-height plot below) while the Vd of TN at Ze > 20 dBZ are all grouped in a narrow Vd range between 0 to -1 m/s

To find out where TP, TN, FP, FN occur, time-height plots of masks of these error matrix scores for the Nov 17-18, 2014 case study are shown in Fig. 3 and the Cloudnet target classification from 0-4 km is shown in Fig 4 while the comparison of Cloudnet and ANN target classification from 0-4 km is shown in Fig 5.

[Figure]

Fig3: Time-height plot of mask of TP, TN, FP, FN for Nov 17-18, 2014 case study using the linear-1 threshold. Shown are the masks up to an altitude of 4 km.

[Figure]

Fig 4: Zoom of Cloudnet target classification in 0-4 km altitude for Nov 17-18, 2014 case study.

[Figure]

Fig 5: Zoom of liquid detection comparison of Cloudnet and ANN (using linear-1 thresholds)  for 0-4 km altitude for Nov 17-18, 2014 case study. Ceilometer cloud base is indicated by black dots.

Fig 3. shows that FN often occur at the outlines of layers of TP, while many FP and TN are related to pixels classified by Cloudnet as rain/drizzle. A closer look showed that: Profiles with low precipitation rates of rain/drizzle have a negative Cloudnet rain flag and are thus not excluded from the analysis. For these drizzle/rain pixel (see Fig 4 between 0-1.5 km), the ANN often predicts liquid. Since the ANN does not distinguish between different liquid classes such as drizzle/rain and cloud droplets (CD), the ANN classifies all these pixels as CD which are then counted as FP. However, in many cases, the ANN does not predict CD when Cloudnet classifies drizzle/rain also leading to high rates of TN in 0-1.5 km as shown in Fig 5.

FN often occur during precipitating profiles which Cloudnet treats as follows: On Nov 17, 2014 3-4 UTC, for example, the lidar detects a very thin liquid cloud in about 700m altitude. Cloudnet classifies all pixel above this first CBH as CD and does thus not distinguish between CD and precipitation falling from the cloud in 1.5 km which is questionable. These drizzle/rain pixel (falsely classified by Cloudnet as CD) which are not predicted to be CD by the ANN are then counted as FN even though the ANN correctly did not classify them as CD. The same issue is evident on Nov 18 when the near-surface fog is present above which's base Cloudnet often classifies entire profiles of CD even though the cloud in 1.5 km produces precipitation. In sum, FN often occur when Cloudnet classifies a certain hydrometeor and extends this target class to the cloud top (also true for ice & supercooled droplets on Nov 17, 17:30 UTC, 1.5-2.5 km).

→ We included part of these explanations in Section 3 of the manuscript where the case Nov 17-18, 2014 case study results are discussed (please see "diff-pdf file").

b) Additionally, we checked if we can find a measure to distinguish between stratiform and convective clouds and thereby evaluate the performance of the hydrometeor target classification.
For that purpose we determined the convective index kappa which characterizes the variability of mean Doppler velocity (Vd) within a time window of 20 min as done in Mosimann, 1995 (https://doi.org/10.1016/0169-8095(94)00050-N) and Kneifel and Moisseev, 2020 (https://journals.ametsoc.org/view/journals/atsc/77/10/jasD200007.xml):

kappa = |Vd(z) - mean(Vd(z))|/mean(Vd(z)). Positive kappa values refer to updrafts, negative values to downdrafts. In Fig 6 below, pixels exceeding kappa values of the range of [-0.5,1] that refer to more turbulent conditions are labeled in black.

The inherent problem of the convective index kappa in trying to distinguish between stratiform/convective conditions to assess in which situations the ANN-based liquid prediction performs well is shown: While turbulent regions near the top of the deep cloud on Nov 18, 3-12 UTC in 7-8 km are correctly flagged, most of the **stratiform** cloud regions in 2-4 km throughout the case study where both, Cloudnet and the ANN classify liquid are marked by high kappa indices. In conclusion, the convective index kappa which is a turbulent measure based on the variance of Vd is thus found to **not** be a good measure to find out where the ANN performs well in detecting liquid and where not as it flags the majority of TP of liquid detection.

[Figure]

Fig 6: Case study of Nov 17-18, 2014 with radar spectrum width (upper panel) and convective index kappa (lower panel) .

We here refrain from further exploration of the performance of the ANN in different turbulence conditions or to separate the performance of the ANN for stratiform vs convective clouds because a) the error matrix scores can often be explained by shortcomings/deficiencies of the ANN itself (it classifies only liquid and does not distinguish between cloud droplets and drizzle/rain) or of the Cloudnet algorithm (e.g. the extension of hydrometeor class found in lower layers to cloud top which e.g. leads to wrong classifications in multi-layer situations with precipitation falling from an upper cloud into a lower cloud) and b) using a convective measure as the convective index kappa was not found to be useful as it would remove the majority of TP.

Rev1: Also, one explanation for the FP of ANN is enhanced SW.To my understanding, the enhanced SW should smear the liquid signature, thus leading to FN. So, turbulence can lead to FP and FN. In what conditions can those two 'bad' classes be expected?

We thank the reviewer for this remark which led us to include a paragraph on the causes of enhanced spectrum width in the manuscript:

The co-existence of multiple hydrometeor types with sufficiently different fall velocities in the same radar volume leads to multimodal Doppler spectra with a high total spectrum width. If the slow-falling hydrometeor has a low reflectivity and narrow peak width, the ANN likely predicts liquid. If there are indeed small cloud droplets and larger ice crystals in the volume, this results in TP. If however there is a co-existence of multiple ice crystal types of which one is small and has a small fall velocity, this results in FP. If however the enhanced SW is not caused by multiple hydrometeor types but by turbulence, liquid peak signatures can be smeared thus leading to FN. In calm conditions (low turbulence) it is more likely that a bimodal spectrum with two ice classes is misclassified as one ice- and one liquid class leading to FP. This problem diminishes with increasing turbulence because of broadening of the peaks and smearing of the individual peaks. The latter (smearing) is the same mechanism for FN in high turbulent conditions.
However, only looking at one variable, the SW, is not sufficient as it is always a combination of MDV, Ze, SW that leads to correct or incorrect classification of liquid class. As shown in the FoO of the radar moments of the error matrix components above, high SW (in combination with low Vd and small Ze) is mostly caused by the TP.

2) Figure 2a. I am curious how well the ANN can predict β and δ. This may also be a part of the 'evaluation'. The accuracy of estimated β and δ may be as important as the selection of thresholds as presented in Table 1. Have you compared the predicted values with observations? At least with beta observed by the ceilometer.

For the MPACE dataset, a comparison between the 2D histograms of observed vs predicted β and δ has been made. Observed (left) and predicted (right) β and δ cluster in the upper left of the 2D histogram. The grey edge encompasses the thresholds for liquid used in Luke et al., 2010.

[Figure]

A comparison of ANN-derived and measured backscatter coefficient and particle depolarization ratio is not directly possible for ACCEPT and thus also not the scope for the underlying study. The ANN retrieval is trained to simulate HSRL particle backscatter coefficient. The lidar instruments operated during ACCEPT provide only attenuated backscatter and volume depolarization ratio. Conversion of attenuated backscatter to particle backscatter requires knowledge about the transmission loss (optical depth) of lidar signal between the instrument (ground) and the range under study. The detected linear depolarization in liquid-water clouds, in turn, relies strongly on the field-of-view of the lidar instrument. With increasing field of view, multiple-scattering effects increase which go along with an increase of the linear volume depolarization ratio (Hogan et al., 2008). Whereas the ACCEPT lidar instruments PollyXT and Ceilometer have fields of view of > 1mrad, the HSRL has a field of view of 0.04 mrad. Therefore, we presume for our study that the ANN retrieval is also valid for non-convective cloud situations as observed during ACCEPT. This is plausible, since the technical specifications of the Mira-35 and ARM cloud-radars (field of view, averaging times) are similar.

Hogan et al., 2008: https://journals.ametsoc.org/view/journals/atsc/65/12/2008jas2642.1.xml

For a visual impression of predicted BSC and depolarization ratio, we are here showing the time-height plots of ANN-predicted values for the Nov 17-18, 2014 case study.

[Figure]

3) P5 L6.'Thirdly, ANN liquid predictions for regions with good lidar echo and Cloudnet-classified as non-liquid class, are reclassified as non-liquid.' This step confuses me. I think it is of importance to know at what conditions the ANN misclassifies lidar-detected non-liquid to liquid. I would not simply ignore this scenario.
HKL: We apologize for the confusion. We tried this early on but do NOT do that step anymore. We deleted the corresponding sentence in the manuscript.

**Minor comments:**

Figure 1. Numbers for subfigures are missing.
Subfigure labeling is now included in Figure 1.

P2 L23 D^6 comes from Rayleigh approximation, which may not be valid for a large fraction of large ice crystals for a cloud radar
True. We added "for the size range in which the Rayleigh approximation is valid".

Figure 2 and 3. I suggest overlap the temperature isothermal lines which will greatly help the interpretation of the results.
True. We now included isotherms.

Figure 3. The green circles are hardly identifiable from black cloud edges. Please use the color which is more contrasting with others.
We added ceilometer CBH as black dots in Figure 1(A-C) and Figure 2 (B) and Figure 3.

Figure3. The overview of this precipitation event has already been presented on Figure 2 (b).I suggest the use of smaller yaxis range. Most interesting signatures are below 2 km. ok
The current yaxis scale seems too large to see the differences among these subfigures are difficult to recognize.The liquid layer above 4 km may deserve a separate figure.
We decided to leave the 0-10 km y-axis range to show the full cloud scene from surface to highest cloud top.Including the ceilometer first cloud base as small black dots should help to improve the clarity of the figure.
We included a zoomed time-height plot (0-4 km) in the Appendix.

Figure3. (Although I doubt the reasonability of 'Thirdly, ANN liquid predictions for regions with good lidar echo and Cloudnet-classified as non-liquid class, are reclassified as non-liquid.') The region marked by red circle should correspond to 'good lidar echo' in Figure 1. Why ANN still identified liquid in this region?
As mentioned above, we do not re-classify the ANN as non-liquid when there is good lidar echo and Cloudnet classified non-liquid. Again, sorry about the confusion.

P9 L22. 'cloud-top layer at −10 °C during 0-6 UTC'. I am confused by this sentence. -10°Cis around 3.5 km. The cloud top during 0-6 UTC Nov18 is definitely much higher. Do you mean 21-24UTC Nov 17?
Sorry, of course. We changed that in the text.

P9 L25.This is interesting. Turbulence favors liquid formation,but may lead to weakened liquid spectral signature if liquid is present. As shown in Figure1, it is obvious that the SW is enhanced at this layer. However, given the weak signal in deBoer2009 and the rather low temperature, it is very unlikely that they are liquid layers. Could you please present examples of the radarDoppler spectrum in this layer as well as at 8 km 6 UTC Nov 18?

Thanks to the reviewer, we realized that in Fig3, we had provided the results before the post-processing step of removing pixel that were classified as liquid at T < -38°C (like at 8km at 6 UTC on Nov 18, 2014). We corrected this in the new version of the manuscript.

Time spectrograms and height spectrograms of Doppler velocities are presented in the figure below.

[Figure]

As illustrated, the Doppler spectrum width during 10-13 UTC on Nov 18, 2014 is high in altitudes of 6.6 - 7 km and smaller above and below this altitude range.  As shown in the range- and height spectrograms, Doppler velocities are usually negative and fluctuate between -0.1 - 0.8 m/s but a few updrafts of up to a few tenths of cm/s are present. Total reflectivies in that time-height range (6.6-7 km 10-12 UTC) are relatively uniform (-20 +/-7 [dBZ]). The Doppler spectra are monomodal. Since ice was formed at higher altitudes in the cloud which then grew by water vapor deposition on its way downward and the spectrum is monomodal we can conclude that NO liquid formation happened at this altitude range at around -37°C - otherwise we would see a bimodal spectrum. The ANN thus most likely misclassified ice as liquid because the observed Doppler spectra were characterized by high spectrum width, small Ze and small Vd. Since the lidar is fully attenuated, we cannot validate this conclusion though.

To check another point of view, we looked into wind profile data. Wind shear can lead to supercooled liquid formation. We thus consulted GDAS1 wind data which however showed only weak wind shear in the altitudes of question thus corroborating the conclusion that no liquid was present at around 7 km altitude.

```
        HGTS  TEMP  UWND  VWND  WWND  RELH     TPOT  WDIR  WSPD
           m    oC   m/s   m/s  mb/h     %       oK   deg   m/s
1000    94.2   7.4  -4.9  -2.3   1.5  86.0    280.5  64.3   5.4
 975     303   5.3  -6.1  -1.8   1.5  95.0    280.5  73.2   6.4
 950     513   3.9  -7.4  -1.3   1.5   100    281.1  79.6   7.5
 925     731   5.0  -7.9  0.65   1.5  91.0    284.4  94.7   7.9
 900     955   3.8  -8.1  0.65   1.5  92.0    285.4  94.6   8.2
 850    1417  0.59  -8.6   1.1     0  98.0    286.8  97.6   8.7
 800    1901  -2.6  -9.1  0.65  -3.8   100    288.4  94.1   9.2
 750    2411  -4.8  -9.1     0  -3.8  86.0    291.3  90.0   9.1
 700    2952  -6.9  -8.9 -0.35  -3.8  56.0    294.8  87.7   8.9
 650    3526 -10.2  -8.8  0.60  -5.1  41.0    297.4  93.9   8.8
 600    4137 -14.9  -9.5   3.4  -3.5  33.0    298.9 109.6  10.1
 550    4788 -20.0 -12.1   5.7  -6.3  33.0    300.4 115.2  13.4
 500    5487 -25.6 -17.2   5.3  -6.9  50.0    301.8 107.2  18.0
 450    6240 -32.5 -18.8   6.3  -7.5  85.0    302.4 108.5  19.8
 400    7060 -38.1 -14.3   7.6 -13.1   100    305.4 117.9  16.2
 350    7964 -45.4 -14.3  11.5 -11.2   100    307.6 128.8  18.4
 300    8972 -54.4 -12.7  17.4  -1.7   100    308.6 143.8  21.5
 250   10124 -55.8  -4.8   7.0     0  82.0    323.2 145.8   8.5
 200   11559 -52.7   1.8  -3.6     0  21.0    349.3 334.1   4.0
 150   13407 -55.5   2.3  -5.5     0  12.0    374.5 336.9   6.0
 100   15969 -58.1   5.6  -8.8  0.22   4.0    415.6 327.6  10.5
  50   20273 -64.9   6.7  -8.7                490.6 322.6  11.0
  20   25779 -67.6  22.8  -8.7                629.4 290.9  24.4
```

Figure 4. The rain flag is missing.

True. We included the Cloudnet rain flag as stated in the figure caption and presented all results in one figure (instead of four subfigures).

P10 L1. May not be the 'Misclassification'. In some cases, e.g. after 7 UTC Nov 18, lidar signals are totally attenuated by the lowest liquid. So, ANN may be correct in the upper layer. Please carefully address this point.

True, we need to phrase this better: Instead of saying on Nov 18 without giving a time restraint, we need to narrow down the time periods where ANN misclassification are likely based on the MWR-LWP comparison.

We rephrased the passage to:

"In some situations the ANN and in others Cloudnet matches the time series of MWR-LWP better. A large discrepancy between ANN-LLT and MWR-LWP is obvious on Nov 18, 4-6 UTC: MWR-LWP are very low, while the ANN-LLT is high. A misclassification of ice as liquid by the ANN in 2-3.5 km height can thus be concluded which is corroborated by the PollyXT lidar signal showing high depolarization values. After 7 UTC on Nov 18, the lidar signals are totally attenuated at the ground and are not available for assessment of ANN classifications in higher layers. Analysis of radar Doppler spectra time- and height spectrograms in around 6-9 km altitude showed only monomodal spectra related to the falling ice. Most certainly, no formation of supercooled liquid in 7 km altitude at -37 °C occurred. "

To further consider this point of how to validate the ANN results in upper cloud layers where the ground-based lidar is fully attenuated, we decided to check if nearby CALIPSO overpasses happened during the ACCEPT field experiment when multi-layer cloud situations were present. As a unique case study of a CALIPSO overpass in 41 km distance, we are now also including the case study of Oct 5, 2014 in the manuscript as Section 3.2. while statistical results for the entire ACCEPT field experiment are moved to Section 3.3.:

"As previously mentioned, no validation of the ANN-liquid prediction can be made if the ground-based lidar signals are fully attenuated. We therefore use the unique opportunity to compare the Cloudnet and ANN liquid identifications in multi-layer cloud situations to a nearby (47 km distant) CALIPSO overpass on Oct 5, 2014 01:05 UTC.

On Oct 5, 2014 01-04 UTC multiple cloud layers were present. Besides warm stratiform liquid clouds below 3 km altitude, a midlevel cloud with cloud top temperature of - 14 C was observed in 5 km altitude. An extensive cirrus was present between 7-10.5 km altitude. From 01- 03 UTC, the PollyXT lidar signal was mostly fully attenuated by the lowest liquid cloud in 1 km altitude leading to a misclassification of liquid as ice by Cloudnet for the warm cloud in 2.5km altitude. Also, (except for a few pixels where the lidar had a valid signal) Cloudnet classified the midlevel cloud as ice-cloud. The ANN correctly predicted liquid for all warm clouds (note that below cloud base of the lowest cloud layer, ANN also predicts liquid which are counted as cloud droplets (CD) since it does not distinguish between different liquid classes such as cloud droplets and rain/drizzle). The ANN classifies the midlevel cloud as liquid-topped with ice precipitating from it below. The phase classification of the ANN in the cirrus is correctly ice except for at the cloud base where high spectrum width and near-zero Vd led to a prediction of liquid.

The cloud fields were extensive (as seen in the MODIS image from midday, which shows the cloudband further east as it was advected from west to east over the past hours) so CALIPSO identified a very similar cloud situation with a cirrus of high vertical extent and a midlevel cloud in 3.5-5 km. The CALIOP signal was fully attenuated in this cloud layer so the low level warm clouds were missed by the satellite observation. The CALIPSO cloud phase index classified the high cloud as ice cloud and the midlevel cloud as liquid-topped cloud with liquid-only or liquid+ice in the lower regions of this cloud. CALIPSO thus validates the ANN-based liquid prediction for the midlevel cloud. This hints to the usefulness of employing satellite-based hydrometeor target classifications as independent validation tool."

[Figure]

[Figure]

ACCEPT Oct 5, 2014 Case Study with CALIPSO overpass for validation

P10 L4.'by comparing the predictions to valid Cloudnet liquid detections'. Do you mean the cloudnet product with 'good lidar echo'?Or regardless of the lidar signal quality?
As previously explained, valid pxl are the union of time-height cells with reliable radar and lidar signal status.

P11 L7. To my understanding, high $\rho_{ceilo-CBH,LLH}$ for ceilometer-CBH and Cloudnet is expected, because cloudnet uses ceilometer data as input. How is this linked to the sensitivity between lidar and radar? I am confused by the logic.
As this paragraph led to several questions, we rephrased it in the answer to major comment 1).

P11 L9.How the averaging affects the performance?

We removed this sentence (as the ceilometer also has a time resolution of 30s) and added a more detailed explanation in the answer to major comment 1.
While the liquid layer base height variable in Cloudnet is based on the gradient of ceilometer backscatter coefficient, the cloud base determination of the ceilometer is not precisely documented in the ceilometer manual. Differences may thus occur due to different methods.

P13L10.The first point may explain the difference between radiosonde and cloud/lidar method, but not the reason why the liquid pixel is higher in cloudnet than ANN.

We assume with "why the liquid pixel is higher in Cloudnet than the ANN" the reviewer refers to the higher overlap fraction of liquid pixels with RH>90% for Cloudnet. We are subsequently explaining more in detail how the differences can be explained:
Not all elements of the error matrix are represented in the overlap fraction of pixel with liquid-detection and RH > 90%: While liquid pixels unrecognized by Cloudnet (i.e. beyond lidar attenuation) are not included in the overlap fraction, wrongly detected ANN liquid pixels (i.e. false positives, FP) are included and thus reduce the fraction of overlap pixel for ANN-predicted liquid.

P14L15. It would be nice to refer to the relevant machine learning techniques. For example, the work by the authors (Kalesse et al., AMT, 2019).
...as well as radar Doppler spectra peak-separation techniques such as PEAKO (Kalesse et al., AMT 2019) and peaktree (Radenz et al., AMT 2019) to check for possibilities of liquid occurrence.

**Typos:**
P9 L25. Nov 18
P11 L7. The high value of rho_ceilo-CBH,LLH…is expected, because…
P11 L21. Case;resulted

Typos have been corrected.

######################################################################

**Reviewer #2:**

The paper investigates the value of using Doppler radar to infer the presence of supercooled liquid layers in clouds. They test the performance of an ANN approach proposed by Luke et al. 2010 on a completely different dataset. As such the study is interesting because it tries to establish how ``portable'' these methodologies are when moving to different cloud regimes. However the paper lacks further analysis and therefore needs substantial improvements before being published. See suggestions below.

**Major comments:**

1) The authors state that ``The objective of this study was to check the performance of the ANN trained with the MPACE observations in Luke et al. (2010)''. Isn't this objective a little bit too limited? For instance the conclusion that the algorithm does not work in convection is pretty obvious given the fact that the Luke's algorithm was trained in low turbulent conditions. But how if we train a ANN in convective regions? Is it more successful or still we have issues because of the intrinsic problem of convection (i.e. smearing of cloud peak)? How do the statistical metrics improve overall?

We agree this would be a good study to perform but it is not our objective here. However, as mentioned in the manuscript, the goal is really to see the portability of the ANN without re-training.

2) I find the description of the four metrics at the end of page 8 a little bit confused. If I follow your guidelines: if precision < 1==> CD overestimation and if recall < 1==> CD underestimation then in all your cases precision and recall are lower than 1 which makes no sense. Please rephrase properly. Same for ``were classified correctly in an absolute and non-balanced way. (overall accuracy)'' not sure what you mean with ``non-balanced'' and what is the meaning of "overall accuracy in the bracket?"

[Figure]

Source: WIKI

The graph above shows the distribution of the two classes predicted by the ANN: CD and non-CD. The beige window-pane like layer represents the decision boundary of the ANN, which can be rotated or shifted in position, depending on the lidar thresholds used. This is to illustrate that the number of FN and FP can grow or shrink independently of each other. Precision and recall therefore take independently different values between 0 and 1, where 1 (perfect) and 0 (bad). If precision becomes smaller, there is more often a "false alarm" (1 - prec = false alarm rate). The recall or "probability for detection" score indicates how often the ANN misses cloud droplets, ergo both are possible: many/few false alarms (CD overestimation) and many/few undetected CD (CD underestimation).

The term "non-balanced" refers to number of pxl of liquid within the entire data set (more ice than liquid is present). We removed the F1-score (it is used to compare different ANN with each other). We removed the word "overall".

Maybe mention that "recall" is the same as ``probability of detection'' (which is a terminology used as well outside the ANN) (right?) and precision =1-false alarm rate (right?).

In the explanation of the scores, we have added "probability of detection" as synonym for recall. And true, precision = 1 - false alarm rate.

Finally it is not clear to me why the authors have not adopted variables more commonly used in literature (like ETS) to assess the overall performances.

We are aware that there are other scores used for verification of forecasts like the Threat Score (TS) or the Equitable-Threat-Score (ETS) (e.g. threat-score) (https://www.cawcr.gov.au/projects/verification/verif_web_page.html):

Below:  hits = TP, misses = FN, false alarms = FP, total = TP + FP + FN + TN

Threat-Score (TS):
TS = hits / (hits + misses + false alarms)
TS answers the question: How well did the forecast "yes" events correspond to the observed "yes" events?
Range: 0 to 1, 0 indicates no skill. Perfect score: 1.
Characteristics: Measures the fraction of observed and/or forecast events that were correctly predicted. It can be thought of as the accuracy when correct negatives have been removed from consideration, that is, TS is only concerned with forecasts that count. Sensitive to hits, penalizes both misses and false alarms. Does not distinguish source of forecast error. Depends on climatological frequency of events (poorer scores for rarer events) since some hits can occur purely due to random chance. TS does not take randomness of the classifyer into account.

Equitable-Threat-Score (ETS) or Gilbert skill score:
ETS = (hits – hits_random) / (hits + misses + false alarms – hits_random), where
hits_random = (hits + misses)*(hits + false alarms)/total

ETS answers the question: How well did the forecast "yes" events correspond to the observed "yes" events (accounting for hits due to chance)?
Range: -1/3 to 1, 0 indicates no skill.   Perfect score: 1.
Characteristics: Measures the fraction of observed and/or forecast events that were correctly predicted, adjusted for hits associated with random chance (for example, it is easier to correctly forecast rain occurrence in a wet climate than in a dry climate). The ETS is often used in the verification of rainfall in NWP models because its "equitability" allows scores to be compared more fairly across different regimes. Sensitive to hits. Because it penalises both misses and false alarms in the same way, it does not distinguish the source of forecast error.

 The Equitable Thread Score is actually not equitable see Hogan et al., 2010: https://journals.ametsoc.org/view/journals/wefo/25/2/2009waf2222350_1.xml

For our two case studies, the ETS achieves lower scores compared to the TS probably caused by the deficiency of Cloudnets' target classification (i.e. liquid extension to cloud top).

Nov17-18 case: TS = 0.588, ETS = 0.432

Oct5-Nov18 dataset: TS = 0.505, ETS = 0.346

As our paper is not about comparing different scores, we will refrain from introducing TS and ETS  in the manuscript.

3) Conclusions 2 and 3 in the abstract are not really corroborated by proper statistical analysis. Which figures/tables prove these statements? Of course we expect such results but we need to prove them.

After adding a more in-depth analysis on the ANN performance based on PDF of radar moments for the error matrix elements in the appendix (see also answer to questions raised by reviewer 1) and after adding another case study where the ANN performance was validated by the CALIOP phase retrieval, we feel that we sufficiently corroborate conclusion 2. We decided to rephrase conclusion 3 and leave out conclusion 4 (since we did not discuss it in depth. The corresponding section of the abstract now is:

"Three conclusions were drawn from the investigation: First, it was found that the threshold selection criteria of liquid-related lidar backscatter and depolarization alone control the liquid detection considerably. Second, all threshold values used in the ANN-framework were found to outperform the Cloudnet target classification for deep or multi-layer cloud situations where the lidar signal is fully attenuated within low liquid layers and the cloud radar is able to detect the microphysical fingerprint of liquid in higher cloud layers. Third, if lidar data is available, Cloudnet is at least as good as the ANN. The times when Cloudnet outperforms the ANN in liquid detections are often associated with situations where cloud dynamics smear the imprint of cloud microphysics on the radar Doppler spectra."

4) Apart from the homogenization step to create more coherent liquid layer structures the overall concept underpinning the ANN methodology is still based on point measurements. My understanding is that there is more potential in these ANN techniques if we try to exploit local information (e.g. involving vertical and horizontal gradients, especially at cloud top e.g. Silber et al, IEEE 2019; Kalogeras et al., 2021 Remote Sensing) and not simply pixel variables (Doppler spectra). But this is not explored at all here because we are still using the approach from 2010 (not much novelty). I see the merit of the current study but the authors should discuss in more depth the way forward.

We agree that citing the mentioned recent publications is of merit to show ways forward and have added the following section in the outlook:

"Additionally, two recent studies also showed the benefit distinguishing between cloud-top liquid-bearing layers and embedded liquid layers when assessing the performance of liquid-detection retrievals (Silber et al., IEEE 2020; Kalogeras et al., Remote Sensing 2021). Silber et al., IEEE 2020 retrieved cloud thermodynamic phase of Arctic clouds based on one year zenith-pointing Ka-band radar and HSRL observations. They found that cloud-top liquid-bearing samples can be more reliably detected than embedded liquid layers as the latter are more difficult to separate from falling ice signatures in the PDF and CDF (cumulative distribution functions) of the first three radar moments as well as Doppler spectra left slope and right slope. Kalogeras et al., 2021 developed a Ka-band radar-only, moment-based technique for supercooled liquid water detection in Arctic mixed-phase clouds. The novelty of this method is that it is a neighborhood-dependent algorithm employing gradients of moments. They concluded that best skill levels for liquid detection are realized for combinations of spectral width and reflectivity vertical gradient and also found their algorithm to be most reliable when applied to cloud tops."

**Minor comments:**

p4, Line 19-21: these instruments have not been used afterwards. I do not see the reason of including them here.

These instruments were only included for completion as to acknowledge the full instrument setup of ACCEPT.

Fig4: Caption: ``Green and red dots near the bottom of the plots" I cannot see them, where are they????

Sorry, rain flag is now added.